# A1CF Binding to the p65 Interaction Site on NKRF Decreased IFN-β Expression and p65 Phosphorylation (Ser536) in Renal Carcinoma Cells

**DOI:** 10.3390/ijms25073576

**Published:** 2024-03-22

**Authors:** Yamin Liu, Jieru Yang, Dunchu Weng, Yajun Xie

**Affiliations:** The Ministry of Education Key Laboratory of Laboratory Medical Diagnostics, The College of Laboratory Medicine, Chongqing Medical University, Chongqing 400016, China; liuyamin2013@126.com (Y.L.); 2021110570@stu.cqmu.edu.cn (J.Y.); 2021110567@stu.cqmu.edu.cn (D.W.)

**Keywords:** A1CF, NKRF, p65(Ser536), IFN-β, renal cancer

## Abstract

Apobec-1 complementation factor (A1CF) functions as an RNA-binding cofactor for APO-BEC1-mediated C-to-U conversion during RNA editing and as a hepatocyte-specific regulator in the alternative pre-mRNA splicing of metabolic enzymes. Its role in RNA editing has not been clearly established. Western blot, co-immunoprecipitation (Co-IP), immunofluorescence (IF), methyl thiazolyl tetrazolium (MTT), and 5-ethynyl-2′-deoxyuridine (EdU) assays were used to examine the role of A1CF beyond RNA editing in renal carcinoma cells. We demonstrated that A1CF interacts with NKRF, independent of RNA and DNA, without affecting its expression or nuclear translocation; however, it modulates p65(Ser536) phosphorylation and IFN-β levels. Truncation of A1CF or deletion on NKRF revealed that the RRM1 domain of A1CF and the p65 binding motif of NKRF are required for their interaction. Deletion of RRM1 on A1CF abrogates NKRF binding, and the decrease in IFN-β expression and p65(Ser536) phosphorylation was induced by A1CF. Moreover, full-length A1CF, but not an RRM1 deletion mutant, promoted cell proliferation in renal carcinoma cells. Perturbation of A1CF levels in renal carcinoma cells altered anchorage-independent growth and tumor progression in nude mice. Moreover, p65(Ser536) phosphorylation and IFN-β expression were lower, but ki67 was higher in A1CF-overexpressing tumor tissues of a xenograft mouse model. Notably, primary and metastatic samples from renal cancer patients exhibited high A1CF expression, low p65(Ser536) phosphorylation, and decreased IFN-β levels in renal carcinoma tissues compared with the corresponding paracancerous tissues. Our results indicate that A1CF-decreased p65(Ser536) phosphorylation and IFN-β levels may be caused by A1CF competitive binding to the p65-combined site on NKRF and demonstrate the direct binding of A1CF independent of RNA or DNA in signal pathway regulation and tumor promotion in renal carcinoma cells.

## 1. Introduction

Apobec-1 complementation factor (A1CF) is an RNA-binding protein that was first described as an APOBEC1 component of the APOBEC1-dependent C-to-U conversion during apolipoprotein B (ApoB) mRNA editing. This process generates ApoB100 and truncated ApoB48 proteins, which have different physiological functions in the regulation of metabolism [1,2]. The role of A1CF in complementing APOBEC1′s RNA editing capacity in vitro is well-defined [1,2,3,4]. A1CF contains three RNA recognition motifs (RRMs) and one putative double-stranded RNA binding domain, which determine the substrate specificity of A1CF by identifying an 11 nt mooring sequence on edited transcripts [5]. However, there is no change in either the editing efficiency of known mRNA edited by the APOBEC1 complex or total ApoB abundance in the liver and small intestine where A1CF is highly expressed. This indicates that A1CF is not essential for APOBEC1-mediated C-to-U editing in vivo under normal physiological conditions [6]. The RNA editing complementation activity of A1CF can be completely substituted by another cofactor of APO-BEC1-RBM47 in vivo [6,7]. A1CF was recently described as a hepatocyte-specific regulator of alternative pre-mRNA splicing (AS), which regulates the AS of two key metabolic enzymes, including ketohexokinase (KHK) and glycerol kinase (GK), as well as other liver-enriched transcripts to influence hepatic lipogenesis and glucose production. A1CF-deficient mice show improved glucose tolerance and are protected from fructose-induced hyperglycemia, hepatic steatosis, and obesity [8].

Earlier studies demonstrated embryonic lethality in germline *A1cf*^−/−^ mice on embryonic day 3.5 because of implantation failure [9]. More recently, another *Sox2*-cre-driven global deletion of *A1cf* in mice did not result in embryonic lethality because they bypassed the implantation stage [10]. In addition to the liver and small intestine, A1CF is highly expressed in the kidney [1,10,11] and has been genetically [12] and molecularly [13] associated with kidney function. Further phenotypic analysis indicates that A1CF loss results in water homeostasis defects and increases protein and solute concentrations in the kidney [10]. Moreover, Blanc et al. reported that A1CF modulates the stability of IL-6 mRNA in hepatic cells by binding to an AU-rich region in the IL-6 3′-untranslated region [14]. A1CF functions in specific mRNA nucleocytoplasmic shuttling, which is dependent upon insulin [15] or interaction with APOBEC1 [16]. Our previous studies have indicated that A1CF regulates cell growth by targeting the 3′-UTR of *Dkk1* mRNA [14,17] and is involved in many cellular processes, such as epithelial-mesenchymal transition (EMT) [13], cell proliferation [18], migration, and apoptosis [17] in proximal tubular epithelial cells and breast cancer cells.

Based on our previous screening of potential A1CF-interacting proteins by Co-IP coupled with Mass Spectrum (MS), we identified a well-known transcriptional silencing factor NF-κB-repressing form factor (NKRF). NKRF specifically inhibits the basal activity of several downstream regulators in the NF-κB pathway, such as interferon β (*IFN-β*), interleukin-8(*IL-8*), and inducible nitric oxide synthase (*iNOS*), by direct binding to their NRE sites [19,20]. Nuclear factor-κB (NF-κB) plays an important role in cell growth [21], inflammation [22], and multiple cancers. p65 is the most important subunit of the NF-κB family [23], and its phosphorylation is involved in the post-translational activation mechanism [24,25].

NF-κB primarily acts as an oncogene, and it has a complex role in tumorigenesis. It is activated in a variety of tumors, including hematologic malignancies and solid tumors [26]; however, the activation of NF-κB induces the expression of proapoptotic genes such as *TRAIL*, *FAS (CD95)*, *PUMA*, *DR4* and *DR5* to promote apoptosis and reduce tumorigenesis [27,28,29]. Bu et al. reported that a p65 phosphomimetic mutant at Ser536 triggers vast apoptosis of colon, breast, and prostate cancer cells and suppresses tumor growth in nude mice [30]. In contrast, the phosphorylation of p65 at Ser536 activates Akt/mTOR signaling to drive malignant hepatocellular proliferation, resulting in hepatocellular carcinogenesis [31]. Therefore, it is important to differentiate between the active sites of p65 and the molecular mechanisms that are attenuated in various cancers for the selective use of NF-κB inhibitors. The primary phenotype of *A1cf*–null mice has been reported in the kidney [10]. Our findings demonstrate the tumor-promoting role of A1CF through the downregulation of p65 phosphorylation at Ser536 and competitive binding to the p65 interaction site on NKRF in kidney-derived cells.

In this study, we confirmed the interaction between A1CF and NKRF and found that the RRM1 of A1CF and the p65 binding site on NKRF are necessary for their interaction. Overexpression of A1CF, but not the RRM1-deleted mutant, decreased the phosphorylation of p65(Ser536) and IFN-β levels, whereas silencing A1CF resulted in the opposite effects. Perturbation of A1CF levels in renal carcinoma cells altered anchorage-independent growth and tumor progression in nude mice. Primary and metastatic samples from renal cancer patients exhibited high A1CF expression and low p-p65 (Ser536) and IFN-β levels in renal carcinoma tissues compared with the corresponding paracancerous tissues. Our results indicate an outside role for A1CF in RNA editing, protein binding independent of RNA or DNA in NF-κB signal pathway regulation, and promoting tumor characteristics in renal carcinoma cells.

## 2. Results

### 2.1. A1CF Interacts with NKRF Independent of RNA and DNA

To determine the functions of A1CF beyond RNA editing, we overexpressed GFP-fused A1CF or GFP in HEK293T cells. Cell extracts were mixed with antibodies against GFP after 48 h of transfection. The immunoprecipitated products were separated by SDS-PAGE and identified after removing the heavy and light chains of the antibody by mass spectrometry. Compared with the control, we found that NKRF was significantly enriched in A1CF-overexpressing cell lysates, and the interaction between A1CF and NKRF was confirmed by co-immunoprecipitation (Figure 1A). Because A1CF and NKRF both contain DNA and RNA binding domains [10,32,33], to exclude the interaction mediated by DNA or RNA, we incubated the cell lysates with DNase and RNase, respectively, before co-IP [34]. DNase or RNase treatment did not weaken their interaction, rather an enhanced interaction was observed with the addition of RNase (Figure 1B), indicating that A1CF and NKRF presented a protein–protein interaction. Immunofluorescence imaging revealed that A1CF colocalized with NKRF in the nuclei using two independent endogenous antibodies, which further confirmed the interaction between A1CF and NKRF (Figure 1C).

### 2.2. RRM Domains of A1CF Bind to NKRF and Competitively Inhibit the Interaction between NKRF and p65

To map the domain function for the A1CF–NKRF interaction, we generated Flag-fused truncated forms of A1CF, including the Flag-RRM domain of A1CF (1-293aa) and Flag-auxiliary domain of A1CF (294-586aa) (Figure 2A). Immunoprecipitation assays revealed that the RRM domains (1-293aa) rather than the auxiliary domain of A1CF, interacted with NKRF (Figure 2B). To determine which RRM domain primarily mediates the interaction, we generated Flag-fused versions of A1CF that were devoid of the RRM1-3 domain (∆RRM1, ∆RRM2, ∆RRM3) (Figure 2A). Immunoprecipitation analysis revealed that RRM1 domain deletion nearly failed to bind to NKRF, whereas RRM2 or RRM3 deletion only attenuated the A1CF–NKRF interaction compared with full-length A1CF (Figure 2C). This suggests that the RRM1 domain of A1CF plays a major role in the A1CF–NKRF interaction.

NKRF acts as an NF-κB repressor protein by directly binding to the p65 subunit [19]. We further prepared GFP-tagged p65 binding motif-deleted NKRF constructs (Figure 2A) and found that deletion of the p65 binding motif of NKRF reduced the interaction with A1CF compared with full-length NKRF (Figure 2D). This indicates that A1CF may participate in NF-κB signaling pathway regulation by modulating the p65 and NKFR interaction or their regulators. To determine the effect of A1CF on the p65 and NKRF interaction, we immunoprecipitated p65 with a specific antibody in the cell lysates of A1CF-overexpressing HEK293T cells and found that A1CF inhibited the p65 and NKRF interaction compared with the control (Figure 2E). This indicates that A1CF negatively regulates the NF-κB signaling pathway by competing with p65 for the same binding site on NKRF. Taken together, the results indicate that the RRM1 domain of A1CF combines with the p65 binding motif on NKRF and competitively inhibits the interaction between NKRF and p65.

### 2.3. A1CF Promotes the Proliferation of Renal Cancer Cells and Decreases the Phosphorylation of p65 (S536) and IFN-β Expression

The NF-κB signaling pathway regulates cell proliferation in a variety of cell types [35,36]. Here, we determined the role of A1CF in two renal cancer cell lines (OSRC-2 and 786-O). Overexpression of A1CF promoted proliferation in OSRC-2 (Figure 3A,B) and 786-O cells (Appendix A), whereas A1CF knockdown by shRNA resulted in the opposite effect (Figure 3C,D; Appendix A). Similar results were observed using an MTT assay (Appendix A), which is an alternative method to EdU for measuring cell proliferation. FCM data indicated that overexpression of A1CF decreased apoptosis (Appendix A), whereas A1CF knockdown increased apoptosis (Appendix A) of OSRC-2 and 786-O cells, which is consistent with the reported results in other tumor cells [17,18] or kidney epithelial cells [13].

Because the A1CF and NKRF interaction weakens the binding of p65 with NKRF, we examined the function of A1CF in NF-κB signaling. Whole-cell lysates were prepared from stable renal cancer cells with disrupted A1CF expression, and we detected the upstream and downstream regulators of NKRF and NF-κB. Surprisingly, both upstream and downstream regulators of NF-κB were associated with the presence or absence of A1CF. In the OSRC-2 and 786-O cell lines, A1CF overexpression decreased p65(Ser536) phosphorylation without altering total p65 and p105 levels (Figure 3E,F, right). The phosphorylation of IκBα (S32), an important site involved in activating NF-κB signaling [37], was markedly inhibited by A1CF (Figure 3E,F, right). A1CF knockdown in these two cell lines enhanced the phosphorylation of p65 (Ser536) and IκBα (S32) (Figure 3E,F, left). IFN-β is a typical NF-κB and NKRF transcriptionally regulated gene [19,38]. It functions in cell apoptosis promotion [39] and was negatively regulated by A1CF in OSRC-2 (Figure 3E) and 786-O (Figure 3F) stable cell lines at the protein level. qPCR assays for the OSRC-2 (Figure 3G,H) and 786-O (Appendix A) stable cell lines revealed that A1CF overexpression inhibited the expression of Ifnb1 mRNA without altering the expression of NKRF or Rela mRNA, and vice versa (Figure 3G,H; Appendix A). To identify the most effective shRNA that specifically targets A1CF, we constructed two shRNAs for A1CF. The results indicated that shRNA#1 exhibited superior silence efficiency (Appendix A), and it was used to screen A1CF-silenced stable cells in renal cancer cells. These results indicate that A1CF inhibits NF-κB activation by disrupting p65 (S536) phosphorylation and negatively regulating IFN-β mRNA levels to suppress the growth of renal cancer cells.

### 2.4. A1CF Inactivates NF-κB by Inhibiting p65, Phosphorylated-p65, and IFN-β Accumulation in the Nucleus

The nuclear translocation of p65 plays an important role in extracellular stimuli that induce the nuclear and transcriptional activation of specific target genes. Upon stimulation, the nuclear localization signal on p65 is exposed and promotes p65 translocation to the nucleus, where it activates transcription factors and induces specific gene expression [25]. Evidence suggests that p-p65(Ser536) is required for p65 activation and nuclear translocation in most cell types [40]. As shown in Figure 3E,F, A1CF negatively regulated p65(Ser536) phosphorylation and IFN-β in cell lysates. To determine the effect of A1CF on phosphorylated-p65(Ser536) distribution in the cytoplasm and nucleus, we prepared cytoplasmic and nuclear extracts of the OSRC-2 and 786-O renal cancer stable cell lines with altered A1CF levels. We found that silencing A1CF promoted the nuclear distribution of p65, phosphorylated p65(Ser536), and IFN-β without altering their distribution in the cytoplasm (Figure 4A). Regardless of the location in the cytoplasm or nucleus, perturbation of A1CF levels had no impact on p105 and NKRF distribution (Figure 4A). Conversely, A1CF overexpression decreased p65 (Ser536) phosphorylation, inhibited p65 nuclear translocation, and reduced IFN-β accumulation in the nucleus without changing the cytoplasmic–nuclear distribution of p105 and NKRF (Figure 4B). Similar results were observed in the 786-O stable cell lines (Figure 4C,D). These results suggest that A1CF inactivates NF-κB by inhibiting p65 and phosphorylated p65 nuclear localization and IFN-β accumulation in renal cancer cells.

### 2.5. A1CF-NKRF Interaction Modulates the Anchorage-Independent Growth through p65 and IFN-β in Renal Carcinoma Cells

Proliferating cancer cells can form a three-dimensional sphere in soft agar, a process known as anchorage-independent growth [41,42]. To determine whether A1CF promotes anchor-independent growth of renal cancer cells, we performed soft agar assays and found that A1CF-overexpressing cells significantly increased the number of colonies compared with that of the control (Figure 5A,B). A1CF-silenced cells showed a decrease in the number of colonies compared with that of the control (Figure 5C,D). To determine the function of the A1CF–NKRF interaction in anchorage-independent growth of renal cancer cells, we overexpressed full-length A1CF or RRM deletion mutants separately (Figure 2A). Only the A1CF-∆RRM1 mutant decreased colony formation, whereas full-length A1CF, A1CF-∆RRM2, and A1CF-∆RRM3 weakened formation without statistical significance (Figure 5E). Co-overexpression of A1CF and NKRF, but not A1CF and NKRF∆(204-308), decreased the A1CF-enhanced anchorage-independent growth of renal cancer cells (Figure 5F). Overexpression of IFNB1 in A1CF-overexpressing 786-O stable cell lines also inhibited A1CF-promoted anchorage-independent growth (Figure 5G).

To clarify the physiological function of the A1CF–NKRF interaction in renal cancer cells, we transfected WT NKRF or NKRF∆(204-308) into A1CF-overexpressing 786-O stable cell lines. The cell lysates were immunoblotted with the indicated antibodies (Figure 6A). Co-expression of A1CF and WT NKRF, but not A1CF and NKRF∆(204-308), restored the decrease in IFN-β protein expression and p65 S536 phosphorylation in A1CF-overexpressing stable cells (Figure 6A), which is consistent with A1CF-driven renal cancer cell proliferation detected by the MTT assay (Appendix A). The A1CF RRM deletion mutants, particularly A1CF ∆RRM1, nearly rescued the reduction in p65(Ser536) phosphorylation and IFN-β expression (Figure 6B) and abolished A1CF-promoted renal cancer cell proliferation as measured by the EdU (Appendix A) and MTT (Appendix A) assays. Correspondingly, overexpression of IFNB1 in A1CF-overexpressing 786-O stable cell lines rescued phosphorylation of p65(Ser 536) (Figure 6C) and cell proliferation as measured by the EdU (Appendix A) and MTT (Appendix A) assays.

The results of anchorage-independent growth and Western blot analysis in renal cancer cells indicated that the A1CF and NKRF interaction modulates anchorage-independent growth through p65 and IFN-β in renal carcinoma cells.

### 2.6. Oncogenic Promoting Characteristics of A1CF in a Xenografted Mouse Model and Renal Tumor Tissues

Based on the above results, we conclude that A1CF promotes cell proliferation and anchorage-independent growth in renal carcinoma cells and that the A1CF–NKRF interaction results in the reduction in phosphorylated-p65(S536) and IFNβ in the nucleus, which plays an important role in these processes. However, it is unclear whether A1CF promotes tumor growth in vivo. Therefore, we injected A1CF-overexpressing OSRC-2 stable cells into the oxter of nude mice, and the tumor volumes of the resulting xenografted mice were measured. Tumors were removed from the mice after three weeks, and A1CF, IFN-β, p65, p65(S536), and ki67 were analyzed by immunohistochemical staining. The results indicated that tumors in the A1CF-overexpressing group grew significantly faster compared with those in the control group (Figure 7A,B). Increased A1CF and ki67, and decreased IFN-β and p65 pS536 were observed in the A1CF-overexpressing OSRC-2 stable cell injection group (Figure 7C). This indicates that A1CF decreases IFN-β expression and phosphorylation of p65(Ser536) in vivo, which is consistent with the in vitro results.

To confirm that the A1CF–NKRF interaction decreases phosphorylated-p65(S536) and IFNβ in the nucleus of clinical samples, we analyzed A1CF expression in renal cancer tissue and adjacent normal tissues from renal cancer patients by immunohistochemistry (Figure 7D). A1CF was highly expressed in renal cancer tissues compared with adjacent normal tissues (Figure 7D). The intensity of IFN-β and p65(S536) phosphorylation were similar to that in the xenografted mouse model, and the expression of p65 was similar in both renal cancer tissues and adjacent normal tissues (Figure 7D).

Collectively, the results of the xenografted mouse model and clinical sample analysis indicated that A1CF possessed oncogenic promotion characteristics and that A1CF–NKRF interaction resulted in the reduction in phosphorylated-p65(S536) and IFNβ nuclear protein levels, which may be the cause of renal carcinoma genesis.

## 3. Discussion

The incidence of kidney cancer continues to increase by approximately 1% annually in both men and women, of which renal cell carcinoma (RCC) is the most common subtype and has the highest mortality rate [43,44]. Early RCC is difficult to detect because it is relatively asymptomatic. Approximately 30% of RCC patients present with metastatic disease at the time of diagnosis, and nearly half of the other patients will subsequently develop metastatic cancer [45]. With the development of imaging techniques and targeted therapy, the survival rate for renal cancer patients has significantly improved. Systemic therapy for RCC, including inhibition of the VEGF signaling pathway using VEGFR tyrosine kinase inhibitors (VEGFR TKIs) and anti-VEGF-Aantibodies, inhibition of the mTOR signaling pathway, and immune checkpoint inhibitors, have shown remarkable efficacy in patients with metastatic RCC [46,47,48]. Furthermore, the development of cancer stem cell targeted therapy and cell therapy for RCC has provided some unique therapeutic approaches [49,50]; activation of the complement system is also an important factor in the development of renal cell carcinoma [51]. However, adverse effects and the development of drug resistance are inevitable [52,53]. Therefore, optimizing inhibitor selection for individual patients with RCC remains a significant challenge.

In this study, we identified Apobec-1 complementation factor (A1CF) as a potential therapeutic target for RCC. A1CF has been widely accepted as an RNA editing protein since its discovery, and its role in complementing the APOBEC1-mediated RNA complex has been studied extensively in vitro. In the present study, we demonstrated an additional role for A1CF beyond RNA editing by screening potential interaction proteins. Our findings indicate that A1CF decreases phosphorylated p65 (Ser536) and IFNβ accumulation in the nucleus by competitively inhibiting the interaction of p65 and NKRF. A1CF binds to NKRF without affecting NKRF expression and nuclear translocation and is independent of DNA and RNA. The RRM1 domain of A1CF and the p65-binding motif of NKRF function in combination. Moreover, we revealed the tumor-promoting effects and NF-κB regulatory function of A1CF in renal carcinoma cells.

Three RRM domains and a putative double-stranded RNA-binding domain in A1CF suggest that its function is strongly associated with RNA [10,54]. As expected, A1CF was identified as a cofactor of the APOBEC1-mediated C-to-U change complex [7]. The mechanism of A1CF in complement to APOBEC1 has been clearly demonstrated in vitro [1,2,3,4,5]; however, recent publications indicated that global ablation of A1CF in mice had no impact on the C-to-U change, particularly in A1CF-highly expressing small intestine and liver, and the A1CF complement function can be completely substituted by RBM47 in vivo [6,7,10]. These reports contrast with previous results, such as embryonic lethality [9] and the APOBEC1 component [1,2,3,4,5]. Furthermore, A1CF has been reported as an important regulator of AS of two key metabolic enzymes, including ketohexokinase (KHK) and GK, and other liver-enriched transcripts in the liver [8]. A1CF is also highly expressed in the kidney and is associated with its physiological function [10,12,13]; however, the mechanism of kidney filtration defects in A1cf-null males and the different phenotypes compared with Apobec1 knockout mice remains unclear [10]. These reports indicate that A1CF may have functions other than RNA editing. Our study is the first to demonstrate the signaling pathway regulation and oncogenic characteristics of A1CF in renal carcinoma cells, thus expanding its biological function.

NF-κB is a complex protein that is ubiquitously expressed in almost all animal cell types. The complex is primarily composed of five members, including p50, p52, p65, c-Rel, and RelB, and dysregulation of this complex results in cancer and immune-associated diseases [25,55,56]. Normally, NF-κB is localized to the cytoplasm as an inactive complex through physical association with inhibitory proteins known as IκBs, which mask the nuclear localization signals (NLSs) of p65 and maintain the p65/p50 heterodimer sequestered in the cytoplasm with its ankyrin repeat domains in an unexposed state, thereby preventing their nuclear translocation and subsequent DNA binding [25,56]. Degradation of IκBα is rapidly initiated following its phosphorylation by an activated IKK complex containing IKKα, IKKβ, and IKKγ. IKKβ is both necessary and sufficient for phosphorylation of IκBα on Ser32 and Ser36 and of IκBβ on Ser19 and Ser23. Phosphorylation of conserved serine residues in IκB proteins results in K48-linked polyubiquitination [57]. Upon stimulation, NLS in p65 is exposed, followed by the phosphorylation and degradation of IκBα, which results in constant shuttling of p65 between the nucleus and cytoplasm. NF-κB is active in a variety of cancers, and the phosphorylation of p65 plays a critical role in tumorigenesis [24,25,26,58]. ZHX2 was reported as an oncoprotein that promotes p65 nucleus translocation without affecting p65 phosphorylation in ccRCC, and the depletion of p65 decreases cell proliferation and anchorage-independent growth [59]; however, the role of p65 phosphorylation in RCC remains unclear.

Many studies have demonstrated the tumor-suppressive role of NF-κB. Overexpression of p65 significantly decreases tumorigenesis in nude mice [28]. NF-κB has been found to induce apoptosis by promoting the expression of proapoptotic proteins [29]. The phosphorylation of p65 in the regulation of NF-κB activity has been extensively studied [60]. At least nine serine and three threonine phosphorylation sites have been identified, including S205, S276, S281, T254, S311, T435, S468, T505, S529, S535, S547, and S536. Phosphorylation at S536 suppresses tumorigenesis by markedly inducing apoptosis [30]. We found that overexpression of A1CF in renal carcinoma cells decreases the phosphorylation of p65 (Ser536) and IkBα (S32) (Figure 3E,F). IKK-mediated phosphorylation of S536 enhances p65 activity by affecting conformational changes in p65 and abrogating its interaction with other proteins [61,62]. In addition, RSK1- or TBK1-mediated phosphorylation of S536 decreases its affinity for IκBα and impairs IκBα-mediated p65 nuclear export [63,64]. These publications indicate that A1CF decreases phosphorylated-p65(Ser536) accumulation in the nucleus and p65 nuclear–cytoplasmic trafficking through a similar mechanism. Further studies are needed to determine whether decreased phosphorylation at S536 of p65 and induction in IFN-β directly suppresses tumorigenesis in RCC and the regulatory mechanism of IkBα(S32) by A1CF.

Based on the literature and our data, we propose a working model of A1CF function in the regulation of NF-κB activity in renal carcinoma cells (Figure 8). A1CF inhibits p65(p-S536) phosphorylation and IFNβ expression and its accumulation in the nucleus depends on the interaction with NKRF, and the A1CF-NKRF-p65/IFNβ signaling axis is important to renal carcinoma growth. This study is the first to demonstrate the oncogenic characteristics and role of A1CF in addition to RNA editing in renal carcinoma cells and may provide additional therapeutic targets for RCC.

## 4. Materials and Methods

### 4.1. Antibodies

For Western blotting assays, the following antibodies were used: A1CF (1:300, ORIGENE, Inc. Rockville, MD, USA, AP50047PU-N), NKRF (1:2000, Proteintech, Inc. Rosemont, IL, USA, 14693-1-AP), IFN-β (1:300, Proteintech, Inc., 27506-1-AP), p65 (1:1000, Cell Signaling Technology, Inc. Danvers, MA, USA, 6956), p50/105 (1:1000, Cell Signaling Technology, Inc., 13586), p-p65 (Ser536) (1:1000, Cell Signaling Technology, Inc., 3033), IκBα (1:1000, Cell Signaling Technology, Inc. 4814), p-IκBα (Ser32) (1:1000, Cell Signaling Technology, Inc., 2859), NPM1 (1:1000, Proteintech, Inc., 60096-1-Ig), β-actin (1:5000, Sigma-Aldrich, Inc. St. Louis, MO, USA, A1978), and GAPDH (1:5000, Sigma-Aldrich, Inc., SAB2701825).

For Immunofluorescence and immunohistochemistry, the following antibodies were used: A1CF (1:50, ORIGENE, Inc., AP50047PU-N), NKRF (1:200, Santa Cruz Biotechnology, Inc. Santa Cruz, CA, USA, sc-365568), IFN-β (1:50, Proteintech, Inc., 27506-1-AP), p65 (1:200, Cell Signaling Technology, Inc., 6956), and p-p65(S536) (1:200, Cell Signaling Technology, Inc., 3033).

### 4.2. DNA Constructs and Mutagenesis

The coding sequence (CDS) of Homo sapiens A1CF (NM_138932), NKRF (NM_001173487), IFNB1(NM_002176.3) were amplified via polymerase chain reaction (PCR) from the cDNA of OSRC-2 cells and cloned at the site of XholI and EcoRI to generate CMVHIS-GFP-A1CF, CMV-Flag-A1CF, CMVHIS-GFP-NKRF, and CMV-HIS-IFNB1 by ligation-independent cloning (LIC) [65]. CMV-Flag-A1CF contained truncated mutations of CMV-Flag-A1CF(1-293aa), CMV-Flag-A1CF (294-586aa) and domain deletion mutations of CMV-Flag-A1CF ∆RRM1, CMV-Flag-A1CF ∆RRM2, and CMV-Flag-A1CF ∆RRM3. CMVHIS-GFP-NKRF contained domain deletion mutations of CMVHIS-GFP-NKRF ∆(204-308).

For A1CF knockdown, two Homo sapiens A1CF short hairpin RNA (shRNA) were cloned into the pLKO.1 vector at sites of EcoRI and AgeI, named A1CF shRNA#1 and A1CF shRNA#2, respectively. All plasmids are listed in Appendix A.

### 4.3. Cell Culture, Transfection, Lentivirus Infection, and Stable Cell Lines

Cells were cultured at 37 °C in a humidified 5% CO_2_ atmosphere. The human clear cancer cell lines 786-O (ATCC CRL-1932) and OSRC-2 (RRID CVCL_1626) were maintained in RPMI-1640 (Sigma-Aldrich, St. Louis, MO, USA) supplemented with 10% fetal bovine serum (FBS), 2 mM glutamine, 1% penicillin–streptomycin (HyClone, Logan, UT, USA). HEK293T cells were cultured in Dulbecco’s modified Eagle’s medium (DMEM) supplemented with 10% fetal bovine serum (FBS), 2 mM glutamine, and 1% penicillin–streptomycin. Cells were seeded at a density of 4 × 10^5^ per 60 mm diameter dish 18 h before transfection. GFP-tagged A1CF overexpression plasmids or sh-A1CF1# silencing plasmids were co-transfected in HEK293T packaging cells with recombined packaging, envelope, and reverse transcriptase vectors with HyFect transfection reagent (Denville Scientific, Holliston, MA, USA) in 10 cm dish. The 786-O and OSRC-2 cells were infected with lentivirus collected 48 h after transfection together with the assistance of polybrene (8 μg/mL) (Sigma-Aldrich, St. Louis, MO, USA). Puromycin (Sigma-Aldrich, St. Louis, MO, USA) was used to select sable single clones. Lipofectamine2000 (Invitrogen, Grand Island, NY, USA) was used for transfecting plasmids (CMV-Flag-A1CF ∆RRM1/CMV-Flag-A1CF ∆RRM2/CMV-Flag-A1CF ∆RRM3/CMVHIS-GFP-NKRF/CMVHIS-GFP-NKRF ∆ (204-308)/CMV-HIS-IFNB1/pLKO.1-A1CF-shRNA 1#/pLKO.1-A1CF-shRNA 2#), which were assessed for the expression of A1CF, A1CF ∆RRM1, A1CF ∆RRM2, A1CF ∆RRM3, NKRF, NKRF ∆ (204-308), and IFNB1 by immunoblots or RT-qPCR after transfection and lentivirus infection.

### 4.4. Immunoprecipitation and Immunoblotting Analysis

Immunoprecipitation and immunoblotting were performed as described previously [66]. HEK293T or 786-O cells were lysed in cell lysis buffer with protease inhibitor mixture (50 mM Tris-HCl pH 7.5, 0.1% SDS, 1% Triton X-100,150 mM NaCl, 1 mM dithiothreitol, 0.5 mM EDTA, 100 mM PMSF, 100 mM leupeptin, 1 mM aprotinin, 100 mM sodium orthovanadate, 100 mM sodium pyrophosphate, and 1 mM sodium fluoride) for 20 min on ice and then clarified by centrifugation at 13,400× *g*. The concentration of protein was measured using the Pierce BCA Protein Assay Kit (Thermo Scientific, Waltham, MA, USA) based on the manufacturer’s instructions, and 1.5mg whole cell lysate was employed for immunoprecipitation with the indicated antibodies. Protein A agarose beads (Millipore Burlington, MA, USA) were incubated with whole-cell lysate at 4 °C for 8–12 h. For DNase/RNase treatment, cell extracts were digested with 50 μg/mL DNase (No. 11284932001, Roche, Basel, Switzerland) and RNase A (No. EN0531, Thermo Fisher Scientific, Waltham, MA, USA) during the incubation with beads. After overnight incubation at 4 °C, immunocomplexes were washed with lysis buffer 3 times and then were analyzed by Western blotting. Samples were applied to SDS-PAGE gels and blotted onto a PVDF membrane (Millipore, IPVH00010). The membranes were then incubated with 5% BSA (Sigma) and primary antibodies for 8 h at 4 °C and then were incubated with HRP-conjugated secondary antibodies for 1 h at room temperature. Signaling was exposed with chemiluminescence, and the band intensity was quantified using the Image Lab software program (v.3.0.0.39296, Bio-Rad, Hercules, CA, USA). The antibodies are listed above.

### 4.5. Immunofluorescence

Immunofluorescence was performed as described previously [3]. Briefly, cells grown on coverslips were fixed with 4% (*w*/*v*) paraformaldehyde (PFA) in PBS and permeabilized with PBS-TRITON 0.5%. After incubation with a blocking solution containing 10% (*v*/*v*) BSA for 1 h to saturate unspecific binding sites, the cells were incubated with indicated primary antibodies, followed by decoration with specific secondary antibodies. Control incubations demonstrated non-cross-reactivity between the anti-Ig conjugates or between the anti-Ig conjugate and the irrelevant primary antibody. Nuclei were stained with DAPI (Sigma-Aldrich). Images were captured with a Leica DM4B microscope (Leica, Wetzlar, Germany). Photographs were taken with a DFC550 Leica camera (Leica, Wetzlar, Germany). The images shown in all figures are representative of at least five random fields (scale bars are indicated).

### 4.6. RNA Extraction and q-PCR

Total RNA was extracted from renal cancer stable cells using TRIzol reagent (Invitrogen, Carlsbad, CA, USA) according to the manufacturer’s instructions. cDNA was prepared by using total RNA (1 μg), oligonucleotide (dT), and random primers in a 20 μL reaction with Revert Aid First Strand cDNA Synthesis kit (Thermo Scientific). One microliter of the cDNA library was used in a 25 μL PCR. Fast SYBR Green Master Mix (Bio-Rad) was used to determine the threshold cycle (Ct) value of each sample using a CFX96 real-time PCR detection system (Bio-Rad). The normalization gene in these studies was 18S. The relative expression levels for the target genes were given by 2DCt (the Ct of 18s minus the Ct of the target gene). Primer sequences used for PCR are listed in Appendix A.

### 4.7. 5-Ethynyl-2′-deoxyuridine (EdU) Assays

First, 5 × 10^3^ stable cells were cultured in a 96-well plate in RPMI-1640 supplemented with 10% fetal bovine serum (FBS), 2 mM glutamine, and 1% penicillin–streptomycin. Then, cell proliferation was detected by the EdU DNA Proliferation Detection Kit (RiboBio, Guangzhou, China) according to the manufacturer’s instructions.

### 4.8. 3-(4,5-Dimethylthiazol-2-yl)-2,5-diphenyltetrazolium (MTT) Assays

Cell proliferation was detected with MTT reagent (Sigma) according to the manufacturer’s instructions. Briefly, 5 × 10^3^ stable cells were cultured in a 96-well plate in RPMI-1640 supplemented with 10% fetal bovine serum (FBS), 2 mM glutamine, and 1% penicillin–streptomycin at different time points (0 h, 24 h, 48, 72 h). The cells were incubated in 200 μL serum-free medium with 5 mg/mL MTT for 4 h. Then, the MTT media was discarded and 150 μL dimethyl sulfoxide (DMSO) was added to dissolve formazan crystals. The absorbance at 490 nm was measured.

### 4.9. Cell Apoptosis Assays

The FITC Annexin-V/Dead Cell Apoptosis Kit with FITC annexin-V and propidium iodide (PI) (Invitrogen, Molecular Probes, Carlsbad, CA, USA) for flow cytometry provides a rapid and convenient assay for apoptosis. Briefly, 1 × 10^5^ stable renal cancer cells were collected and washed twice with PBS and then mixed with 100 μL binding buffer to form a cell suspension. FITC annexin-V 5 μL was mixed with 1 μL of 100 μg/mL PI working solution. The cells were incubated in the dark for 15 min at room temperature. Then, 400 μL binding buffer was added, mixed gently, and analyzed immediately using an FACS.

### 4.10. Nucleus and Plasma Protein Extraction

Nucleus and plasma protein were isolated from renal cell stable cell lines as previously described [67,68]. Briefly, approximately 2 × 10^7^ cells per extract were prepared, and the cells were washed gently with PBS buffer. The cells were lysed for 3 min on ice in 300 μL of Cytoplasmic Extract Buffer (10 mM HEPES, 60 mM KCl, 1 mM EDTA, 0.075% (*v*/*v*) NP40, 1 mM DTT, and 1 mM PMSF, adjusted to pH 7.6) and centrifuged at 1200 rpm for 4 min. The pellet containing intact nuclei was resuspended and gently washed in 100 μL of Cytoplasmic Extract Buffer without detergent. Then, 1 pellet volume of Nuclear Extract buffer (20 mM Tris Cl, 420 mM NaCl, 1.5 mM MgCl_2_, 0.2 mM EDTA, 1 mM PMSF, and 25% (*v*/*v*) glycerol, adjusted to pH 8.0) was added to the nuclear pellet, the salt concentration was adjusted to 400 mM using 5 M NaCl, and an additional pellet volume of NE buffer was readded. The pellet was resuspended by vortex, and the extract was incubated on ice for 10 min to extract nuclear protein. Plasmatic and nuclear extractions were centrifuged at high speed to discard cell membrane debris and then the purified supernatant containing nucleus or plasma protein was analyzed by Western blotting.

### 4.11. Soft Agar Assay

To assess the anchorage-independent colony formation in renal cancer cells expressing different plasmids, 1 × 10^5^ cells were suspended in a complete medium containing 0.3% agar and plated in 6-well plates over a basal layer of a complete medium containing 0.6% agar. Cells were cultured for 2 weeks and then were stained with 0.005% crystal violet. The images of the plates were analyzed using Image J software (v.2.35, National Institutes of Health, Bethesda, MD, USA). Each experiment was set in triplicate, and the statistical analysis was performed using Prism 5 software. The relative number of colony cells was adjusted to control cells.

### 4.12. Tumorigenicity in Nude Mice

Animal assays were approved by the Ethics Committee of Chongqing Medical University. To assess the tumorigenicity in renal cancer cells expressing different plasmids, 1 × 10^7^ OSRC-2 stable control cells or the same number of A1CF overexpression cells were injected hypodermically into the armpit of the right forelimb of BALB/C-nude mice (male, 6 weeks old, five mice per group). Tumor size was measured with a dial caliper, and the mice were euthanized on day 21. The volume of the excised tumors (mm^3^) was calculated using the formula: V = (length × width^2^)/2.

### 4.13. Immunohistochemistry

Tissue sections were fixed with 4% formaldehyde, embedded in paraffin, and cut into 4 mm thick sections. Staining for A1CF, IFN-β, NF-κB p65 pS536, ki67, and total NF-κB p65 was carried out following a standard immunostaining protocol 21. Briefly, the sections were deparaffinized with xylene, dehydrated with ethanol, and then heated in 0.01 M citrate buffer (pH 6.0). Endogenous peroxidase activities were inactivated in 3% H_2_O_2_ for 10 min at room temperature, and the sections were blocked with 3% normal goat serum in 0.2 M PBS (pH 7.4). The samples were then incubated with anti-A1CF, anti-IFN-β, and anti-p65 pS536 at 37 °C for 1 h. Secondary anti-mouse antibody-coated polymer peroxidase complexes were then applied for 30 min at 37 °C, followed by treatment with substrate/chromogen and further incubation for 5–10 min at 37 °C. The slides were counterstained with hematoxylin. Images were captured with a Leica DM4B microscope (Leica, Wetzlar, Germany). Photographs were taken with a DFC550 Leica camera (Leica, Wetzlar, Germany).

### 4.14. Statistical Analysis

All experiments were performed independently three times, and the results are presented as the mean ± standard error of the mean (SEM) or SD. Data were assessed for statistical significance using Student’s *t*-test and repeated measures analyses of variance were conducted by one-way analysis of variance (ANOVA) with Tukey’s post hoc comparisons. GraphPad Prism 5 software (GraphPad, San Diego, CA, USA) was applied to calculate the statistical results. Statistical differences were considered significant with * *p* < 0.05, ** *p* < 0.01, *** *p* < 0.001.

## Figures and Tables

**Figure 1 ijms-25-03576-f001:**
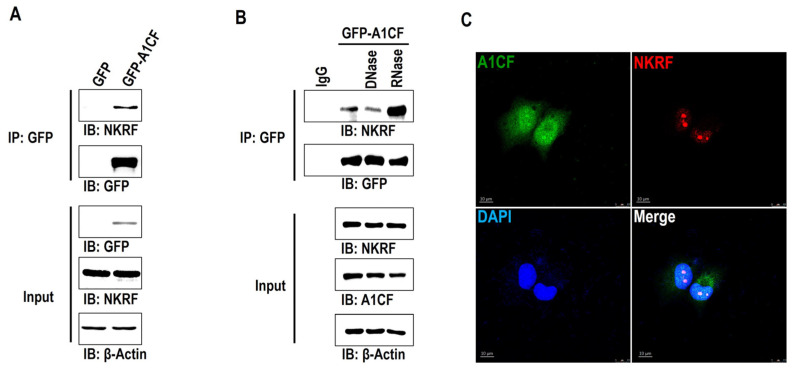
A1CF interacts with NKRF independent of RNA and DNA. (**A**) Extracts from HEK293T cell lines expressing GFP-tagged A1CF or the GFP-tag alone were used for immunoprecipitation with an anti-GFP antibody immobilized on Protein A agarose beads. Inputs and eluates containing co-precipitated proteins were analyzed by Western blotting using antibodies against the indicated proteins. (**B**) A1CF-overexpressed 786-O stable cell extract was used in immunoprecipitation (IP) experiments in the presence of DNase or RNase. Inputs and eluates containing co-precipitated proteins were analyzed by Western blotting using antibodies against NKRF, A1CF, and β-Actin as indicated. (**C**) Immunofluorescence of A1CF and NKRF localization. The 786-O cells on glass coverslips were fixed, probed with antibodies to A1CF and NKRF, and analyzed by immunofluorescence. Images were captured with a Leica DM4B microscope (Leica, Wetzlar, Germany). Photographs were taken with a DFC550 Leica camera (Leica, Wetzlar, Germany). Images shown in all figures are representative of at least five random fields.

**Figure 2 ijms-25-03576-f002:**
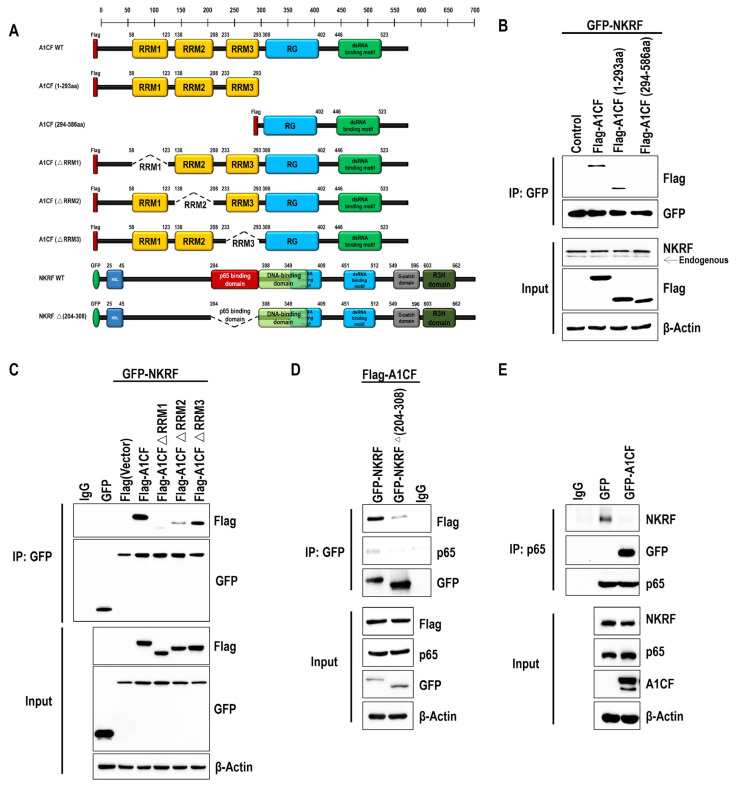
RRM domains of A1CF that bound to NKRF and competitively inhibited the interaction of NKRF and p65. (**A**) Schematic view of the domains of A1CF and NKRF. Amino acid numbers corresponding to the boundaries of specific domains are indicated above. The truncated A1CF Flag-tagged protein and NKRF GFP-tagged protein constructs were used for immunoprecipitation. (**B**) Immunoprecipitation experiments were performed using extracts from GFF-NKRF transfected HEK293T cell lines co-expressing the Flag-tag alone or Flag-tagged versions of A1CF, truncated A1CF (1-293aa), and A1CF (294-586aa). (**C**) HEK293T cells were co-transfected with GFP-NKRF and Flag-tagged A1CF WT or A1CF RRM1-3 mutants as indicated. Immunoprecipitation with anti-GFP antibodies was performed. (**D**) Extracts from HEK293T cell lines co-expressing Flag-tagged A1CF with GFP-tagged NKRF or NKRF ∆(204-308) that lacked the p65 binding domain were used in immunoprecipitation experiments and analyzed by Western blotting. (**E**) The 786-O stable cell lines expressing GFP-tagged A1CF or the GFP-tag alone were used for immunoprecipitation with an anti-p65 antibody immobilized on Protein A agarose beads. Immunoprecipitation and immunoblotting analyses of 786-O cell lysates were performed with the indicated antibodies.

**Figure 3 ijms-25-03576-f003:**
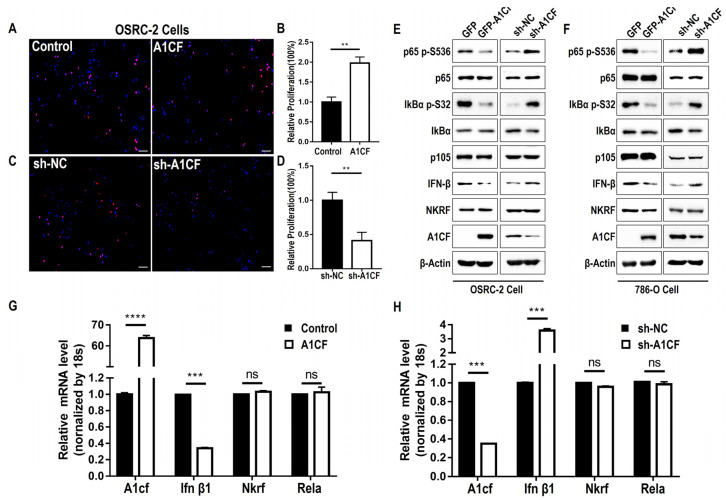
A1CF promoted the proliferation of renal cancer cells and decreased the phosphorylation of p65 (S536) and IFN-β expression. (**A**–**D**) Growth of OSRC-2 stable cells was measured by 5-ethynyl-21-deoxyuridine (EdU) assay. Results were expressed as mean ± SEM of 3 replicate wells. *p*-values were calculated by Student *t*-tests, ** *p* < 0.01. (**A**) Merged image of EdU staining (red) and Hoechst staining (blue) for OSRC-2 A1CF overexpression stable cells. Scale bar, 50 μm. (**C**) Merged image of EdU staining (red) and Hoechst staining (blue) for OSRC-2 A1CF deficiency stable cells. Scale bar, 50 μm. (**B**,**D**) Statistical analysis of OSEC-2 cell proliferation. Values were presented as mean ± SEM (*n* = 3). *p*-values were calculated by Student *t*-tests, ** *p* < 0.01. (**E**) OSRC-2 stable cells were examined by Western blotting with the indicated antibodies. (**F**) The 786-O stable cells were examined by Western blotting with the indicated antibodies. (**G**,**H**) mRNA levels of indicated genes in OSRC-2 stable cells were measured by RT-qPCR. The data are presented as means ± SD from triplicate samples. **** *p* < 0.0001, *** *p* < 0.001, ns means no significance.

**Figure 4 ijms-25-03576-f004:**
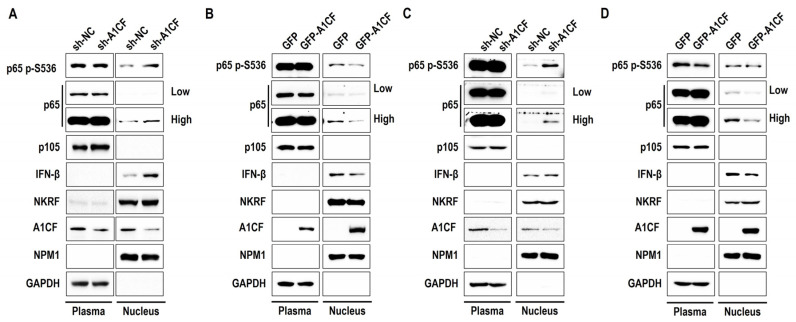
A1CF inactivated NF-κB by inhibition of p65, phosphorylated-p65, and IFN-β accumulation in the nucleus. (**A**) Cytoplasmic and nuclear extracts from OSRC-2 A1CF deficiency cell lines were prepared and subjected to Western blotting analysis with the indicated antibodies. (**B**) Cytoplasmic and nuclear extracts from OSRC-2 A1CF overexpression cell lines were prepared and subjected to Western blotting analysis with the indicated antibodies. (**C**) Cytoplasmic and nuclear extracts from 786-O A1CF deficiency cell lines were prepared and subjected to Western blotting analysis with the indicated antibodies. (**D**) Cytoplasmic and nuclear extracts from 786-O A1CF overexpression cell lines were prepared and subjected to Western blotting analysis with the indicated antibodies.

**Figure 5 ijms-25-03576-f005:**
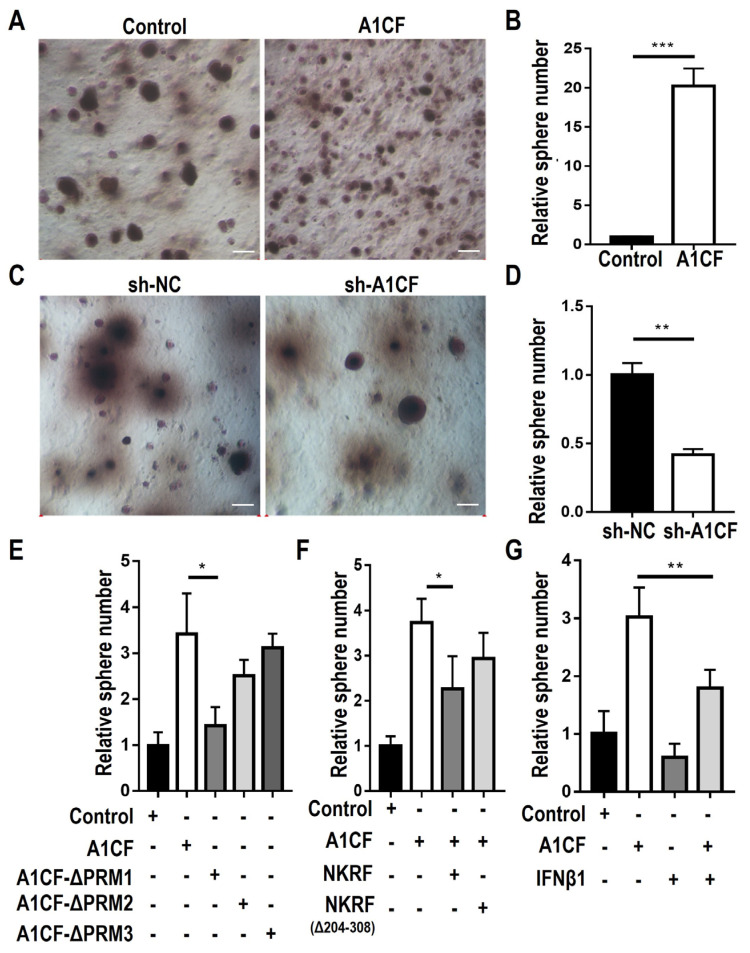
A1CF enhanced the anchorage-independent growth of renal cancer cells dependent on the A1CF-NKRF interaction. (**A**–**D**) The effects of A1CF on soft agar colony formation in human renal cancer cell lines. (**A**) Colony formation assay in 786-O overexpression stable cells. Scale bar, 200 μm. (**C**) Colony formation assays in 786-O deficiency stable cells. Scale bar, 200 μm. (**B**,**D**) Statistical analyses of 786-O colony formation. Quantitative data using Image J software (v.2.35) were presented as mean ± SEM (*n* = 3). *p*-values were calculated by Student *t*-tests, ** *p* < 0.01, *** *p* < 0.001. (**E**–**G**) The statistical analyses of colony formation in 786-O cells expressing the indicated plasmids. Colonies were counted using ImageJ software (v.2.35). Results are presented as the mean ± SEM (*n* = 3). *p*-values were calculated by Student *t*-tests, * *p* < 0.05, ** *p* < 0.01.

**Figure 6 ijms-25-03576-f006:**
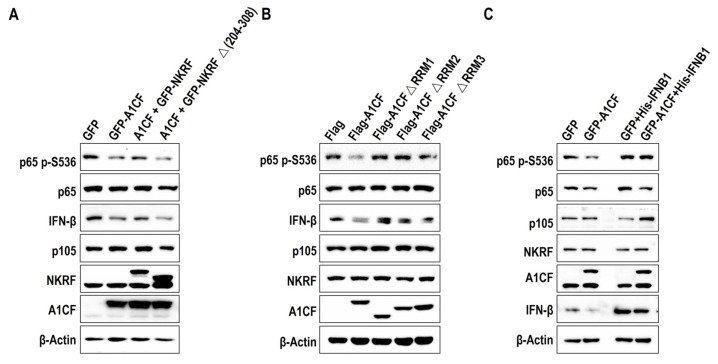
The A1CF-NKRF interaction is necessary for p65(S536) phosphorylation and IFN-β expression. (**A**–**C**) The 786-O cells expressing the indicated plasmids were examined by immunoblotting analyses with the indicated antibodies.

**Figure 7 ijms-25-03576-f007:**
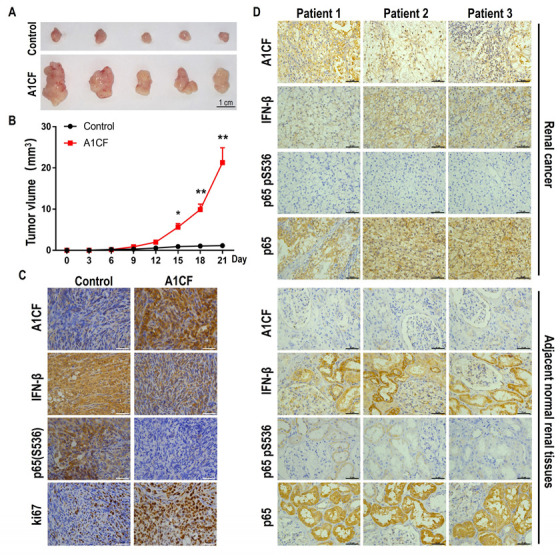
Oncogenic promotion characteristic of A1CF in a xenografted mice model and renal tumor tissues. (**A**) Nude mice were subcutaneously injected in the right flank with OSRC-2 control cells and A1CF overexpression stable cells. An image of excised tumors at 21 days post-injection is shown. (**B**) Xenograft tumor growth curve of control or A1CF-overexpression OSRC-2 stable cells in nude mice. After subcutaneous implantation, the short and long diameters of the tumors were measured and tumor volumes (mm^3^) were calculated every 2 days in mm^3^ (*n* = 5). * *p* < 0.05, ** *p* < 0.01. (**C**) Representative expression levels of A1CF, IFN-β, NF-κB p65 pS536, and ki67 in xenograft tumors were detected by immunohistochemical. Scale bar, 50 μm. (**D**) Representative expression levels of A1CF, IFN-β, NF-κB p65 pS536, and total NF-κB p65 in renal cancer and adjacent normal tissues by immunohistochemistry. Scale bar, 50 μm.

**Figure 8 ijms-25-03576-f008:**
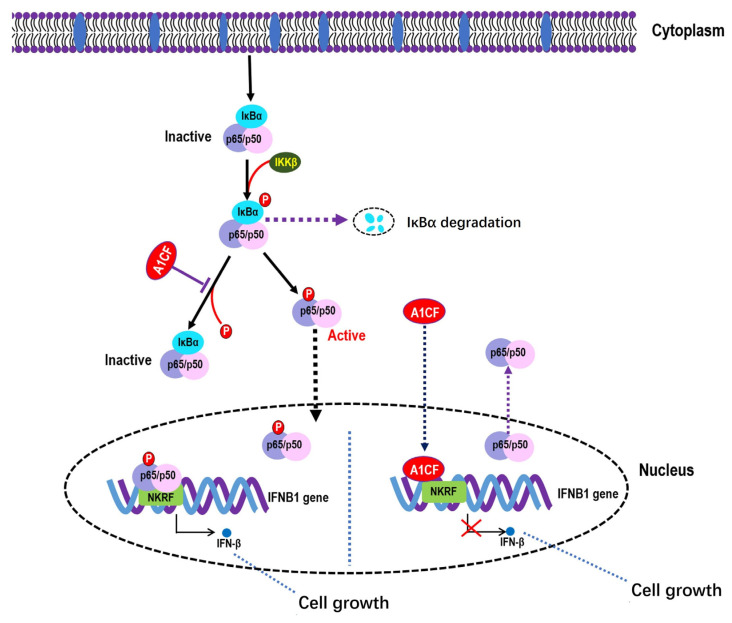
Graphical abstract representing the A1CF regulated cell growth through competitive binding to the p65 interaction site on NKRF-induced decreased IκBα phosphorylation, inhibited NF-κB nuclear translocation, and reduced p65 (Ser536) phosphorylation in renal carcinoma cells.

## Data Availability

All data generated or analyzed during this study are included in this published article and its Appendix A.

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
