# Peer review of "A1CF Binding to the p65 Interaction Site on NKRF Decreased IFN-β Expression and p65 Phosphorylation (Ser536) in Renal Carcinoma Cells"

_ijms, 2024, doi:10.3390/ijms25073576_

Round 1
Reviewer 1 Report
Comments and Suggestions for Authors
The article presents novel findings on A1CF's role in renal carcinoma cells, highlighting its interaction with NKRF and the subsequent modulation of p65 phosphorylation and IFN-β levels, independent of RNA and DNA. This interaction influences cell proliferation and tumor progression, offering new insights into A1CF's functions beyond RNA editing and its potential as a target in cancer therapy.
Novelty and Interest:
- The demonstration of A1CF's direct binding to NKRF, influencing p65 phosphorylation and IFN-β levels independently of its known RNA-binding functions, marks a significant advancement in understanding A1CF's role in cellular signaling and cancer progression.
- The exploration of A1CF's effect on renal carcinoma cells' proliferation and the implications for tumor growth and progression in vivo provide a valuable foundation for further investigation into potential therapeutic targets.
Areas for Improvement:
- Methodological Clarification: More detailed descriptions of experimental conditions and methodologies could enhance reproducibility and the validity of the findings.
- Discussion: The discussion section of the manuscript requires significant enhancement to fully articulate the implications of your findings within the broader scientific context. It should more comprehensively address how your results compare and contrast with existing literature, particularly emphasizing the novel aspects of your study. Furthermore, elaborating on potential limitations, suggesting directions for future research, and discussing the clinical relevance of your work will strengthen the overall impact of your manuscript. Providing a deeper analysis of how your findings contribute to our understanding of renal carcinoma and potential therapeutic approaches could greatly improve the discussion's effectiveness. The identification of cancer stem cells (CSCs) expressing CD133+/CD24+ in clear-cell renal cell carcinoma (ccRCC) underscores the complexity of tumor biology and highlights potential targets for innovative therapies (Reference: PMID: 37685983). The findings on A1CF's interaction with NKRF and its implications for cellular signaling in renal carcinoma cells offer an additional layer of understanding, suggesting that targeting both CSC-related pathways and A1CF-NKRF interactions may present a comprehensive approach for therapeutic development in RCC. Moreover, The emerging role of the complement system in the pathogenesis of renal cell carcinoma (RCC) provides a vital context for our findings on A1CF's impact on renal carcinoma cells. Recent studies, including a narrative review (PMID: 38003705), underscore the complement system's involvement in RCC's TME, suggesting a multifaceted role in tumor progression. This backdrop enriches our discussion on A1CF's potential interactions within the TME, highlighting the importance of considering both innate immune responses and specific molecular interactions in understanding RCC's complexity.
- Functional Assays: Further functional assays to elucidate the precise mechanisms by which A1CF influences p65 phosphorylation and IFN-β expression could clarify its role in NF-κB signaling.
- Broader Implications: Discussion on the broader implications of A1CF's functions in renal carcinoma and potential therapeutic applications would enhance the relevance of the findings.
- Limitations: Addressing potential limitations and biases in the study design and execution could improve the credibility and reliability of the research.
In summary, while the study presents groundbreaking insights into A1CF's role in renal carcinoma cells, enhancing methodological transparency, expanding statistical and comparative analyses, and further exploring the molecular mechanisms and therapeutic implications could make it more robust and impactful for publication.
Reviewer 2 Report
Comments and Suggestions for Authors
This is a very interesting manuscript addressing the role of APOBEC1 Complementation Factor (A1FC) in renal carcinogenesis. While this is a very well-known feature for A1FC, its mechanistic roles have not been very well studied. The authors claim that A1CF interaction with NF-kappa-B-repressing factor (NKRF) determines the reduction of phosphorylated-p65(S536) and IFNβ nuclear protein level, which is turn are responsible for renal tumorigenesis.
While this is important for cancer biology, the authors must clarify few critical things.
The rationale for studying renal cancer is not clearly defined. Using 786-O and OS-RC-2, a very well-known low/negative A1RC cell lines does not have also a rationale. Overexpressing a gene in a cell line can artificially apply to any tumor, including the ones where A1FC was not establishes as an oncogene. Not sure how a gene which anyway inherited is low/negative can be downregulate. This underlines an important thing. Source of cells must be mentioned, including authenticity and mycoplasma assays.
Can this hypothesis be translated in a syngeneic renal tumor model? Is this gene oncogenic in mouse renal tumor models? Can be targeted to treat cancer?
The most intriguing question remains the mechanism behind this oncogene. Does, indeed, reduction of phosphorylated-p65(S536) and IFNβ nuclear protein level translate to pro-carcinogenic effect. It will be fantastic to show (and should as a critical positive control) in a simple experiment if KO IFNβ and impairing p65 phosphorylation, indeed, leads to a pro-carcinogen effect.
As a minor comment, manuscript needs extensive editing (many repetitions, typo, etc.)
Round 2
Reviewer 1 Report
Comments and Suggestions for Authors
The manuscript demonstrates A1CF's interaction with NKRF affecting NF-κB signaling and renal carcinoma progression, suggesting A1CF as a therapeutic target. Supported by in vitro and in vivo data, it merits publication. Ref 12 should be reformatted
Author Response
Dear reviewers,
We are very sorry for our carelessness. After receiving your comments, we changed the format of Ref12 using the reference style of MDPI in EndNote. Ref12 has been modified to:
Huang, L.Y.; Wang, H.L.; Zhou, Y.R.; Ni, D.S.; Hu, Y.X.; Long, Y.S.; Liu, J.N.; Peng, R.; Zhou, L.; Liu, Z.C.; et al. Apobec-1 Complementation Factor (A1CF) Inhibits Epithelial-Mesenchymal Transition and Migration of Normal Rat Kidney Proximal Tubular Epithelial Cells. Int J Mol Sci 2016, 17(2),197.
The changes to our manuscript were also highlighted by using red-colored text. (page23, lines673-675)
Thank you again for your positive comments and valuable suggestions to improve the quality of our manuscript.
Sincerely,
The Authors
18 Mar 2024
Reviewer 2 Report
Comments and Suggestions for Authors
I would like to thank the authors for their detailed answers to reviewer's question. As this research may trigger fundamental questions in RCC carcinogenesis, in the present format the manuscript may be taken in consideration for publication.
One minor thing, which authors missed to answer: Have the cell lines checked for authenticity and periodically mycoplasma assay, since a lot of cells manipulation has been done on some cell lines?
